# *Pentaclethra eetveldeana* Leaves from Four Congo-Brazzaville Regions: Antioxidant Capacity, Anti-Inflammatory Activity and Proportional Accumulation of Phytochemicals

**DOI:** 10.3390/plants12183271

**Published:** 2023-09-15

**Authors:** Victor N’goka, Sandrine Lydie Oyegue Liabagui, Cédric Sima Obiang, Herman Begouabe, Gelase Fredy Nsonde Ntandou, Romeo Karl Imboumy-Limoukou, Jean-Claude Biteghe-Bi-Essone, Brice Serge Kumulungui, Jean Bernard Lekana-Douki, Ange Antoine Abena

**Affiliations:** 1Laboratoire de Pharmacodynamie et de Physiopathologie Expérimentale (L2PE), Faculté des Sciences et Techniques, Université Marien Ngouabi (UMNG), Brazzaville BP 69, Congo; 2Unité d’Evolution, Epidémiologie et Résistances Parasitaires (UNEEREP), Centre Interdisciplinaire et de Recherches Médicales de Franceville (CIRMF), Franceville BP 769, Gabon; 3Département de Biologie, Faculté des Sciences, Université des Sciences et Techniques de Masuku (USTM), Franceville BP 876, Gabon; 4Ecole Doctorale Régionale d’Afrique Centrale en Infectiologie Tropicale (ECODRAC), Université des Sciences et Techniques de Masuku (USTM), Franceville BP 876, Gabon; 5Laboratoire de Recherches en Biochimie (LAREBIO), Faculté des Sciences, Université des Sciences et Techniques de Masuku (USTM), Franceville BP 876, Gabon; 6Département de Parasitologie-Mycologie Médecine Tropicale, Faculté de Médecine, Université des Sciences de la Santé (USS), Libreville BP 4009, Gabon; 7Laboratoire de Biochimie et de Pharmacologie (LBP), Faculté des Sciences de la Santé, Université Marien Ngouabi (UMNG), Brazzaville BP 69, Congo

**Keywords:** *Pentaclethra eetveldeana*, antioxidant, leaf aqueous extract, flavonols, flavones, secondary metabolites, membrane stabilization, antidenaturation

## Abstract

Oxidative stress and inflammation play a key role in the occurrence of neurodegenerative diseases. Traditionally, *Pentaclethra eetveldeana* leaves are used in dementia treatment. Therefore, this study aimed to evaluate the antioxidant and anti-inflammatory activities as well as the phytochemical composition of *Pentaclethra eetveldeana* leaves from four Congo-Brazzaville regions. The 1.2-diphenyl-1-picrylhydrazyl radical-scavenging, β-carotene bleaching and molybdenum reduction assays were used to assess the antioxidant activity. The protein denaturation and erythrocyte membrane stabilization tests were used to analyze the anti-inflammatory activity. Phytochemical screening, the quantification of polyphenols by spectrophotometry, as well as the determination of extraction yields were carried out. It was found that the extracts reduced molybdenum; furthermore, compared to ascorbic acid, they showed better antiradical activity and inhibited lipid peroxidation. Moreover, globally, the membrane-stabilizing power of the aqueous extracts was superior or comparable to diclofenac, while the same extracts were less effective for the inhibition of denaturation. All of the aqueous extracts contained polyphenols, saponins, alkaloids, anthraquinones, reducing sugar and cardiotonic glycosides. The total polyphenols, tannins and proanthocyanidins are produced proportionally from one region to another. Finally, the leaves from Brazzaville and Boundji contain flavonols, while those from Makoua and Owando contain flavones. Thus, *Pentaclethra eetveldeana* leaves contribute to traditional dementia treatment through their antioxidant and anti-inflammatory properties.

## 1. Introduction

Living healthy, supporting the well-being of everyone at all ages, as well as preserving and restoring the earth’s flora by ensuring their sustainable use are two sustainable development goals of this century [1]. Reaching these goals, however, requires a deep knowledge of plant resources, especially their biological activities and their phytochemical compositions. In accordance with this, it is imperative to consider each plant in its different growth environments, in order to better know and understand each of them. Therefore, their use becomes effective or rational by removing doubt regarding the homogeneity or heterogeneity of their biological properties, as with their chemical compositions according to their geographical distribution.

Furthermore, the establishment of these sustainable development goals is undeniable proof that humanity is constantly seeking ways to promote health and well-being [2]. Indeed, the health and well-being of individuals is essential to the socio-economic and political development of nations [3,4]. For this reason, any public health problem is a major concern in all countries of the world. As proof, the COVID-19 pandemic is a concrete and recent example. The evolution of the SARS-CoV-2 infection into severe forms, resulting in disproportionate inflammatory host responses, caused severe disruptions to health systems worldwide, leading to significant socio-economic destabilizations [5,6,7]. On the other hand, the production of reactive oxygen species during the inflammatory process is a normal reaction of the body in response to infection or tissue damage [8]. Consequently, it contributes to the initiation of several biological functions and the restoration of normal physiology [9]. Conversely, hyperinflammation and excessive production of reactive oxygen species are the cause of neurodegenerative, metabolic and cardiovascular diseases, carcinoma, angiogenesis, infertility, etc. [10,11,12,13,14,15,16,17,18].

Considering the major role played by oxidative stress and inflammation during dysfunctions of the body, many investigations have been undertaken. These consisted, on the one hand, of understanding and explaining oxidative stress [19,20] and inflammation [21,22] and, on the other hand, in the search for new antioxidant and anti-inflammatory molecules. In this context, interest has increasingly focused on plants [23,24,25,26], in view of the multiple metabolites that they produce. These metabolites are widely used in the cosmetic, food and pharmaceutical industries [27]. Among these compounds, polyphenols are recognized for their therapeutic properties. For example, polyphenols are proposed as adjunctive treatments for psychiatric and neurodegenerative disorders [28]. Other studies have shown the powerful anxiolytic and antidepressant effects of phenolic extracts from parsley, as well as the neuroprotective effects of catechin from green tea [29,30]. Additionally, polyphenols are known to have anti-inflammatory and antioxidant effects [31]. Furthermore, given their importance for human health, anti-inflammatories and antioxidants still remain the object of chemical syntheses by researchers [32,33]. However, the long-term use of synthetic antioxidants produces carcinogenicity, cytotoxicity, oxidative stress induction and endocrine-disrupting effects [34,35]. Likewise, current anti-inflammatories, despite the benefits they bring, are known for their harmful effects on the body [36,37]. Meanwhile, the reassuring fact is that plant metabolites, by their functioning in synergy, confer advantages to the organism [38]. Therefore, to know them better would be a major asset.

In 2016, Christenhusz et al. estimated the number of plant species to be 374,000 [39]. Among these plants is *Pentaclethra eetveldeana* (*P. eetveldeana*) De Wild & T. Dur. (Fabaceae-Mimosoideae). *P. eetveldeana* is an endemic plant of the Guineo-Congolese region [40]. This plant is frequently mentioned in phytogeographical and botanical studies [41,42,43,44,45,46,47,48,49], while little is known of its phytochemical, pharmacological and toxicological levels. For instance, in the Democratic Republic of Congo, Memvanga et al. reported the antiplasmodial effect of the root bark of this plant [50]. In the same way, triterpenoid saponins have been isolated from its bark [51,52]. In particular, traditionally in the Republic of Congo, *P. eetveldeana* bark is used in the treatment of various respiratory tract diseases (cough, bronchitis, whooping cough, tuberculosis), in intoxications, constipation, stomach pains, hernia outbreaks, but also for genitourinary infections and various generalized pains. Furthermore, decoctions from the bark are also used against filariasis and as an anthelmintic, while in association with several non-mentioned plants, the leaves of *P. eetveldeana* are used in the treatment of dementia, a neurodegenerative disease [53]. In other words, this predisposes *P. eetveldeana* leaves to antioxidant and anti-inflammatory effects in view of the close link between the occurrence of neurodegenerative diseases with oxidative stress and inflammation [10,18].

Thus, this research aimed to explore the potential antioxidant and anti-inflammatory effects as well as the secondary metabolites composition of *P. eetveldeana* leaves from four regions of the Republic of Congo.

## 2. Results

The results were obtained after investigations carried out on four aqueous extracts of *Pentaclethra eetveldeana* leaves collected at 6 a.m. in the department of Brazzaville, from the districts of Boundji, Owando and Makoua. For simplification, the aqueous extracts of *Pentaclethra eetveldeana* leaves from Brazzaville, Boundji, Owando and Makoua will be called AqBRA, AqBOU, AqOWA and AqMAK, respectively.

### 2.1. Extract Yields

The yields were calculated after weighing dry aqueous extracts obtained from 65 g of Pentaclethra eetveldeana dry leaves. The results are reported in Table 1. These results showed that the highest yields were from AqBRA, followed by that of AqBOU. Then, the lowest yield was from AqOWA, while that of AqMAK was in third place.

### 2.2. Phytochemical Composition

#### 2.2.1. Metabolites Identified in *Pentaclethra eetveldeana* Leaf Aqueous Extracts

As shown in Table 2, the phytochemical screening revealed the presence of alkaloids, saponins, reducing sugars, anthraquinones, cardiotonic glycosides and polyphenols in all aqueous extracts, while oses and holosides as well as sterols and triterpenes were only identified in AqBOU and AqBRA. To summarize, on the one hand, based on the reactions of each test, a very positive reaction was observed for all extracts in the case of the saponins, polyphenols and reducing sugars tests. Furthermore, a very positive reaction was also observed with the anthraquinone and cardiotonic glycoside tests for AqOWA and AqMAK, a positive reaction for AqBOU and AqBRA with the cardiotonic glycoside test, and weak and positive reactions for AqBRA and AqBOU, respectively, with the anthraquinone test. Additionally, with the alkaloid test, AqBOU showed a very positive reaction, AqBRA and AqMAK showed a positive reaction, while AqOWA showed a weak reaction. Finally, with the oses and holosides test, both AqBOU and AqBRA showed a very positive reactions, contrary to that with sterols and triterpenes, where AqBOU and AqBRA showed a weak reaction and a positive reaction, respectively.

Moreover, among the phenolic compounds, like shown in Table 3, coumarins, tannins and flavonoids were found in aqueous extracts of the leaves from all regions, while anthocyanins were only found in AqBRA. It is interesting to note that flavonols were identified in AqBOU and AqBRA, while flavones were identified in AqOWA and AqMAK. For more precision, with all of the tests, a very positive reaction was observed for all aqueous extracts except the anthocyanin test, with which AqBRA showed a weak reaction.

#### 2.2.2. Quantities of Phenolic Compounds in *Pentaclethra eetveldeana* Leaf Aqueous Extracts

Figure 1 shows the quantities of total polyphenol (TP), proanthocyanidins (PR), tannins (TN) and flavonoids (FL) expressed in µg of gallic acid equivalent/g (µg GAE/g), µg of apple proanthocyanidin equivalent/g (µg APE/g), µg of tannic acid (µg TAE/g) and µg of quercetin equivalent/g (µg QE/g), respectively.

The TP, PR, TN and FL ranged from 1602.22 ± 61.49 to 348.33 ± 15.43 µg GAE/g; from 2108.22 ± 79.41 to 206.55 ± 23.94 µg APE/g; from 583.18 ± 37.58 to 112.81 ± 33.96 µg TAE/g and from 41.65 ± 1.43 to 16.86 ± 4.01 µg QE/g, respectively. Clearly, in the cases of TP, PR and TN, AqBRA (TP: 1602.22 ± 61.49; TN: 2108.22 ± 79.41; TN: 583.18 ± 37.58) showed the highest concentrations, while AqOWA (TP: 348.33 ± 15.43; TN: 206.55 ± 23.94; TN: 12.81 ± 33.96) showed the lowest concentrations. In addition, those of AqBOU (TP: 1135.55 ± 55.51, PR: 1708.22 ± 12.61, TN: 527.62 ± 54.55) were higher than those of AqMAK (TP: 694.16 ± 22.56, PR: 677.11 ± 10.04, TN: 322.07 ± 15.16). Furthermore, concerning FL, AqMAK (41.65 ± 1.43) showed the highest concentration, followed by AqBOU (34.46 ± 5.33), then AqOWA (26.86 ± 5.90), and AqBRA (16.86 ± 4.01) showed the lowest quantity.

### 2.3. Antioxidant Activity of Aqueous Extracts from the Four Regions

The concentrations of aqueous extracts that scavenge 50 percent of DPPH radicals (IC_50_) ranged from 0.53 to 8.24 µg/mL, and were lower than that of ascorbic acid, as shown in Table 4. Among the extracts, in ascending order, AqBRA followed by AqBOU and then AqMAK showed the lowest IC_50_ values compared to AqOWA.

The results concerning the molybdenum assay that measures total antioxidant capacity, expressed as µg ascorbic acid equivalent/g of extract (µg AAE/g), are reported in Table 5. As reported, AqBOU showed the greatest total antioxidant capacity followed by AqBRA. Moreover, the total antioxidant capacity of AqMAK was higher than that of AqOWA, which was the lowest of all.

Finally, the results of the beta carotene bleaching assay, expressed as relative antioxidant activity (%), are shown in Figure 2. As stated, the relative antioxidant activities (RAA) of all of the aqueous extracts were higher than the RAA of ascorbic acid (100 ± 0.00%) and distilled water (14.10 ± 2.22%). Briefly, with a percentage equal to 124.72 ± 1.45%, AqBRA presented the highest RAA followed by AqBOU (121.79 ± 1.26%). Then, the RAA of AqMAK was 118.13 ± 2.74%, and the lowest was that of AqOWA (116.11 ± 3.74%).

### 2.4. Correlation between Antioxidant Activity, Phenolic Compounds and Extraction Yields

Table 6 shows the correlation coefficients (R^2^) between extraction yields, concentrations of phenolic compounds and antioxidant activity of *Pentaclethra eetveldeana* leaves from Boundji, Brazzaville, Owando and Makoua. The green and red colors represent positive and negative correlation coefficients, respectively, and when these colors become clearer, the correlation coefficients tend to zero. Clearly, with R^2^ = 0.886, as shown in Figure 3, the lowest quantity of flavonoids is related to the highest relative antioxidant activity, except in the case of AqOWA. Moreover, there is also a positive link (R^2^ ranging from 0.796 to 0.972) between extraction yields (EY), total polyphenols (TP), proanthocyanidins (PR), tannins (TN) and total antioxidant capacity (TAC). In short, as shown in Figure 4a, EY, TP, PR, TN and TAC decrease or increase from one region to another when one of them decreases or increases. On the other hand, the DPPH radical scavenging activity (IC_50_ in µg/mL) correlates negatively (R^2^ ranging from −0.866 to −0.678) with EY, TP, TN and PR. Briefly, as illustrated in Figure 4.b, the lowest IC_50_ is associated with the greatest quantity of compounds, and vice versa. Finally, with R^2^ ranging from 0.858 to 0.998, a strong positive correlation was observed between EY and the quantities of TP, PR and TN.

### 2.5. Anti-Inflammatory Activity

#### 2.5.1. Antidenaturation Activity of *Pentaclethra eetveldeana* Leaf Aqueous Extracts

The inhibitory activity of the aqueous extracts on protein denaturation is shown in Figure 5. Clearly, except for AqMAK and AqOWA, the results showed that AqBOU and AqBRA mainly inhibited denaturation at 500 µg/mL, but only AqBOU at 15.625 µg/mL. Moreover, at all of the concentrations tested, the inhibition percentages of extracts were lower than those of diclofenac. At 500 µg/mL, in ascending order, the inhibition percentages of the extracts were 17.96 ± 0.29% (AqOWA), 30.38 ± 0.29% (AqBRA), 44.01 ± 1.18% (AqBOU) and 49.90 ± 0.37% (AqMAK). On the other hand, at 15.625 µg/mL, also in ascending order, the inhibition percentages of denaturation were 42.26 ± 0.08% (AqBOU), 53.34 ± 0.56% (AqMAK) and 65.01 ± 0.17% (AqOWA). Furthermore, it was also observed that from 250 to 15.625 µg/mL, the inhibition percentages of AqOWA increased from 1.29 ± 0.08 to 65.01 ± 0.17%. Finally, in general, for AqMAK and for all the other extracts, the best inhibitions of denaturation were observed at concentrations of 500 and 15.625 µg/mL.

#### 2.5.2. Membrane Stabilization Activity of *Pentaclethra eetveldeana* Leaf Extracts

##### Inhibition of Heat-Induced Hemolysis

The inhibition percentages of heat-induced hemolysis are shown in Figure 6 below. It was observed that all of the aqueous extracts of the Pentaclethra eetveldeana leaves inhibited hemolysis at all concentrations tested. First of all, at 500 µg/mL, the inhibition percentages of all extracts ranged from 63.42 ± 0.11 (AqBRA) to 84.83 ± 0.09% (AqBOU), which were superior to the inhibition percentage of diclofenac (59.81 ± 0.19%). Similarly, at 31.25 µg/mL, except for AqMAK, the inhibition percentages of AqBOU, AqBRA and AqOWA ranged from 81.89 ± 0.08 (AqBOU) to 98.46 ± 0.11 % (AqBRA), and were higher than that of diclofenac (67.35 ± 0.31%). Furthermore, from 500 to 62.5 µg/mL, AqBOU (500 µg/mL: 84.83 ± 0.08%; 250 µg/mL: 73.06 ± 0.11%; 125 µg/mL: 67.33 ± 0.14% and at 62.25 µg/mL: 91.59 ± 0.08%) showed the highest inhibition percentages compared to AqBRA (63.42 ± 0.11; 60.83 ± 6.83; 59.83 ± 0.28 and 72.93 ± 0.28%), AqOWA (81.54 ± 0.20, 60.91 ± 0.26; 51.21 ± 0.23 and 82.23 ± 0.11%) and AqMAK (74.33 ± 1.71; 56.75 ± 2.11; 43.03 ± 0.08 and 40.29 ± 0.62%). In addition, at 31.25 and 15.625 µg/mL, AqBRA (98.46 ± 0.11 and 83.48 ± 0.56%) and AqOWA (96.52 ± 0.54%; 93.60 ± 0.08%) showed the highest inhibitions, while the lowest inhibitions were reported for AqMAK. Finally, the inhibition percentages of AqBOU were 81.89 ± 0.11 and 73.40 ± 0.69 at 31.25 and 15.625 µg/mL, respectively.

##### Inhibition of Hypotonicity-Induced Hemolysis

Similarly, to heat-induced hemolysis, all of the aqueous extracts of the Pentaclethra leaves inhibited hypotonicity-induced hemolysis at all concentrations tested, as reported in Figure 7 below. To summarize, it was observed that all extracts, with some exceptions, showed the highest inhibition percentages compared to diclofenac (500 µg/mL: 24.51 ± 2.44%; 250 µg/mL: 23.08 ± 0.04%; 125 µg/mL: 18.16 ± 0.13%; 62.5 µg/mL: 11.52 ± 0.25%; 31.25 µg/mL: 12.96 ± 0.19%; 15.625 µg/mL: 10.60 ± 0.12%). The exceptions were AqBRA (18.91 ± 0.12%) and AqMAK (19.42 ± 0.06%) at 250 µg/m, AqBRA (14.42 ± 0.12%) at 125 µg/mL, AqBOU (12.50 ± 0.22%) at 31.25 µg/mL and AqBOU (9.57 ± 0.18%) at 15.625 µg/mL. Obviously, these exceptions were also the lowest inhibition percentages among the extracts. Moreover, at 500, 250 and 125 µg/mL, AqBOU (500 µg/mL: 40.31 ± 0.06%; 250 µg/mL: 24.92 ± 0.13%; 125 µg/mL:29.92 ± 0.30%) and AqOWA (500 µg/mL: 35.02 ± 0.08%; 250 µg/mL: 26.92 ± 0.14%; 125 µg/mL: 20.74 ± 1.48%) showed the highest inhibition percentages. On the other hand, at 62.5 and 31.25 µg/mL, AqOWA (19.27 ± 0.12% and 17.33 ± 0.16%) and AqMAK (18.02 ± 0.16% and 17.68 ± 0.19%) presented the highest hemolysis inhibitions; then, at 15.625 µg/mL, it was found that AqBRA (14.03 ± 0.16%) and AqOWA (15.01 ± 0.10%) exhibited the highest inhibition percentages. For the others, the lowest inhibition percentages were those of AqBRA (24.01 ± 0.12%) and AqMAK (25.29 ± 0.25%) at 500 µg/mL, and those of AqBOU (13.94 ± 0.12%) and AqBRA (13.13 ± 0.15%) at 62.5 µg/mL.

## 3. Discussion

In this study, experiments were conducted to assess the antioxidant and anti-inflammatory activities of *Pentaclethra eetveldeana* leaves, in order to demonstrate the role played by these leaves in the traditional treatment of dementia. Concurrently, a phytochemical exploration was carried out to explain the link between biological activities and chemical composition.

The qualitative phytochemical evaluation indicated that all leaves from the four localities contain alkaloids, saponins, cardiotonic glycosides, anthraquinones, reducing sugar and polyphenols. This shows a homogeneity in the types of secondary metabolites produced, due to the fact that all leaves come from the same species [54]. It is well known that plants produce secondary metabolites in response to external and internal changes by having primary metabolites as substrates [54]. Therefore, plants also contain primary metabolites such as sugars. In line with this fact, one study reported the presence of metabolites mentioned below in plant leaves [55]. Moreover, among the phenolic compounds, coumarins, gallic tannins and flavonoids were those that were identified. These compounds are known to be abundant in plant leaves [56]. Concerning the flavonoids group, flavonols were exclusively detected in leaves from Brazzaville and Boundji, while flavones were present in the leaves from the other localities. To support this, Jin et al. reported that both flavones and flavonols can be find in the same part of a plant within the same species [56]. Furthermore, oses and holosides as well as sterols and terpenes were only detected in leaves from Brazzaville and Boundji. This agrees with the β-chemodiversity, which means there are intraspecific qualitative variations due to adaptive responses of the species to the environment [57].

After phytochemical screening, the quantitative analysis revealed extraction yields, then concentrations of total polyphenols, proanthocyanidins, tannins and flavonoids in the aqueous extracts. The results showed that including the extraction yields of TP, PR, TN and FL, there is a variability from one region to another This heterogeneity of yields and phenolic compound concentrations can be explained by biotic and abiotic factors that lead the plant to change its physiology, which impacts the accumulation of secondary metabolites and, more specifically, the availability of nutrients, which vary according to soil types as one of the determining factors [27]. This intraspecific variability in secondary metabolites accumulation was reported by Maffi et al. in more than seven plants of *Hypericum perforatum*, regarding the total flavonoids content in flowers [58]. Furthermore, recently, leaves of *Inula helenium* L. collected in three localities presented variations in total polyphenols and flavonoids [59]. Additionally, in all of the extracts, proanthocyanidins and total polyphenols were more abundant, followed by tannins and then flavonoids. These results agree with those of Ngoua et al., who reported the quantity of phenolic compounds in aqueous extracts of *Lophira procera* [60]. Finally, a correlation was observed between the extraction yields and the concentrations of total polyphenols, proanthocyanidins and tannins. This shows that when one of these quantities decreases or increases from one locality to another, the others also decrease or increase concurrently. This analysis suggests that despite the variability of quantities from one region to another, these compounds are always produced proportionally. In accordance with the fact that the production of secondary metabolites depends on the genotype of the species [54], it is obvious that the production of the secondary metabolites mentioned above, found in all leaf extracts, is included in the genotypes of *Pentaclethra eetveldeana* species.

*Pentaclethra eetveldeana* leaf extracts demonstrate antioxidant activities. First and foremost, based on the fact that DPPH is a stable free radical [61], any substance capable of scavenging it possesses free radical scavenging activity. Secondly, the molybdenum test measures the ability of substances to donate electrons in order to reduce oxidative or intermediate free radicals. Accordingly, a substance possesses reducing power when it reduces molybdenum (IV) into a green-colored molybdenum (V) complex [62]. Therefore, in line with the previous statements, aqueous extracts of *Pentaclethra eetveldeana* leaves possess an antioxidant capacity by reducing molybdenum (V) and scavenging DPPH radicals. This activity is justified by the properties of flavonoids to donate hydrogen atoms or electrons [63]. Moreover, the correlation test showed that both the scavenging activity and the reducing power increase or decrease concurrently with the quantities of compounds. As a result, leaves from Brazzaville followed by those from Boundji and then those from Makoua showed the highest antioxidant activity. In accordance with our study, Ngoua et al. reported antioxidant activities of plant extracts with the same methods [60]. Thirdly, through the β-carotene bleaching test, antioxidant compounds prevent β-carotene from being oxidized by free radicals from linoleic acid [64]. These peroxide-type radicals are generated by heat. In the body, they are responsible for the destruction of DNA and proteins through direct interactions with them [65]. Thus, the aqueous extracts are able to inhibit lipid peroxidation. As mentioned above, this activity is due to the ability of flavonoids to donate hydrogen atoms [63,66]. In addition, all of the aqueous extracts showed higher activity than ascorbic acid. Furthermore, concerning lipid peroxidation inhibition, a positive correlation was observed with flavonoids. Clearly, the lowest quantity corresponds to the greatest activity, except for the aqueous extract of leaves from Owando. This suggests that the relative antioxidant activity is not related to the quantity of flavonoids. Unquestionably, it is well known that antioxidant power is related to the number of hydroxyl groups in the molecular structure [67].

Like in many cases of disease, inflammation plays an important role in neurodegenerative diseases. Firstly, the antidenaturation assay was used to assess the ability of extracts to inhibit protein denaturation. Protein denaturation is caused by the production of autoantigens in the body, leading to severe inflammation [68]. At 15.625 µg/mL, the extracts mostly showed effects greater than 50% but less than diclofenac, which is known for its protective effect on proteins against denaturation [69]. In agreement with our results, William et al. reported that *Boehmeria jamaicensis* and *Glincida sepium* leaf extracts protected proteins against denaturation at concentrations below 2 µg/mL [70]. The possible protection mechanism of aqueous extracts can be related to the ability of flavonoids to bind to proteins, but it was also reported that, after binding with proteins, quercetin and rutin alter their secondary structures [71]. This can explain the fact that at 250–62.5 µg/mL, aqueous extracts do not inhibit protein denaturation. Secondly, the red blood cell membrane stabilization test was chosen to assess the protective effect of the extracts against lysosomal membranes, due to their similarity with red blood cell membranes [72,73]. Indeed, any substance capable of stabilizing the membrane prevents the phenomenon of autophagy. This phenomenon, by destroying the lysosomal membrane, causes the release of lysosomal contents in the organism. Consequently, this leads to the destruction of cells as well as proteins [74,75]. Thus, after the investigation, the aqueous extracts showed a stabilizing effect on hemolysis induced both by hypotonicity and heat. In the case of heat induction, the aqueous extracts showed greater inhibition than diclofenac, mostly at low and high concentrations. Moreover, for induction by hypotonicity, the inhibitions of the extracts were comparable to that of diclofenac, or even greater. These results can be explained by the ability of phenolic compounds such as flavonoids to rigidify the erythrocyte membrane [76,77]. In agreement with our results, authors have also proved that plant extracts stabilize the erythrocyte membrane [60,78].

This research brings new knowledge about *Pentaclethra eetveldeana* leaves, especially regarding their phytochemical composition, antioxidant activities and anti-inflammatory activities. However, further investigations are needed to conduct studies at the molecular level, as well as in vivo studies to accurately determine the origins of the metabolite variations.

## 4. Materials and Methods

### 4.1. Species

*Pentaclethra eetveldeana* leaves (Figure 8) were collected at 6 a.m. in September 2022 in the districts of Makoua, Owando, Boundji and in the department of Brazzaville (Republic of Congo). Therefore, we had a total of four collections. Then, the collected leaves were dried at room temperature and crushed.

### 4.2. Preparation of Extracts

By decoction, the aqueous extracts were prepared by boiling 65 g of dried leaves for 20 min with distilled water. After cooling, the mixtures were filtered, and the filtrate obtained was evaporated in an oven at 60 °C; thus, the aqueous dry extracts were obtained. Finally, the dry extracts were weighed to determine the yields using formula below.
Yield (%) = (mass of dry extract/mass of crushed leaves) × 100(1)

### 4.3. Phytochemical Screening

The primary and secondary metabolites were identified in the aqueous extracts by the various tests reported in Table 7. All tests were carried out with 2 mL of aqueous extract.

### 4.4. Quantification of Phenolic Compounds

#### 4.4.1. Total Polyphenols

The Folin–Ciocalteu method described by Aryal et al. [91] was used, with a few modifications. The mixtures consisted of 0.25 mL of extract (1 mg/mL) with 1.25 mL of 10% (*w*/*v*) Folin–Ciocalteu reagent, followed by the addition of 1 mL of Na_2_CO_3_ (20%) 5 min after. Then, the mixture was incubated for 10 min at room temperature in the dark. After incubation, at 765 nm, the absorbances were measured using a spectrophotometer (Thermo Scientific Genesys 10S UV-VIS. Waltham, MA, USA). Finally, the results were expressed as gallic acid equivalents in µg per gram of dry extract (µg GAE/g).

#### 4.4.2. Tannins

The quantification of tannins was carried out according to the method described by Obame Engonga [92], with a few modifications. A volume of 0.25 mL of extract was mixed with 1.25 mL of water, 0.25 mL of ferric ammonium citrate (28% iron; 3.5 g/L) and 1 mL of aqueous ammonia (15%). Then, the absorbances were measured after 10 min at 525 nm. The results were expressed in µg of tannic acid equivalent per g of dry extract (µg TAE/g).

#### 4.4.3. Proanthocyanidins

Proanthocyanidins were quantified according to the method described by Dicko et al. [93], with some modifications. The reaction mixture was constituted with 0.17 mL of extract and 2.33 mL of hydrochloric acid (30%, in butanol). Then, the reaction mixtures were vortexed and then heated at 100 °C for 2 h. After cooling, the absorbances were measured at 550 nm with a spectrophotometer. The results were expressed in µg equivalent of apple proanthocyanidins per g of dry extract (µg APE/g).

#### 4.4.4. Flavonoids

The AlCl_3_ method described by Quettier-Deleu et al. [94], with a few modifications, was to mix 0.5 mL of extract (1 mg/mL, prepared in methanol) with 0.5 mL of AlCl_3_ (2%, methanol), followed by 10 min of incubation at room temperature. Then, the absorbances were read at 430 nm. The results were expressed as µg quercetin equivalent per g of dry extract (µg QE/g).

### 4.5. Evaluation of Antioxidant Activity

#### 4.5.1. DPPH Radical-Scavenging Assay

The 1.1-diphenyl-2-picrylhydrazyl (DPPH) radical-scavenging activity of extracts was determined through the method described by Yunusa et al. [95], with some modifications. The DPPH solution at 50 µg/mL (*w*/*v*) was prepared by dissolving 10,000 µg of solid DPPH in 200 mL of 100% methanol, and 1 mL of the DPPH solution was mixed with 1 mL of five different concentrations (12.5 to 0.7812 µg/mL) of each extract. Then, the mixture was incubated for 30 min in the dark, and the absorbance were measured at 517 nm. Ascorbic acid was used as a reference antioxidant at the same concentrations mentioned above, with methanol (100%) as a blank. Finally, the concentration that inhibits 50 percent of the radicals (IC_50_) (µg/mL) was calculated for each of the substances tested, using the following equation:Inhibition (%) = (AbsC − AbsS/AbsC) × 100(2)
where AbsC is the absorbance of the mixture with methanol, and AbsS is the absorbance of the mixture with extract or reference antioxidant (ascorbic acid).

#### 4.5.2. Total Antioxidant Capacity: Molybdenum Assay

The total antioxidant capacity was determined by the phosphomolybdenum method as described by Aliyu et al. [96], with some modifications. Briefly, the reaction mixture was composed of 0.3 mL of extracts (1 mg/mL) and 3 mL of the reagent (0.6 M sulfuric acid, 28 mM sodium phosphate and 4 mM ammonium molybdate). The mixtures were then incubated at 70 °C for 90 min in a water bath, followed by a cooling step at room temperature. Then, using a spectrophotometer at 695 nm, the absorbances were measured against the blank. At last, the total antioxidant activity was expressed as milligram equivalent of ascorbic acid/g of extract.

#### 4.5.3. β-Carotene Bleaching Assay

The antioxidant activity was assessed using the β-carotene linoleate model system, with a slight modification, according to Ghedadba et al. [96]. A mass of 0.5 mg of β-carotene was dissolved in 1 mL of chloroform, 50 μl of linoleic acid and 500 μL of Tween-20. After removing chloroform at 50 °C, 100 mL of distilled water was added, and the mixture was vigorously agitated to form an emulsion. Then, 2.5 mL of the emulsion was added into different test tubes containing 0.5 mL of extract or reference antioxidant (ascorbic acid) at 1 mg/mL, and immediately afterwards the mixtures were heated for 2 h in a water bath at 50 °C. The absorbances, at 470 nm, were measured 48 h later from the time the reaction mixtures were complete. The relative antioxidant activity (RAA) was calculated, as indicated below. AbsS is the absorbance of each extract, and AbsAA is the absorbance of ascorbic acid.
RAA (%) = (AbsS_(48h)_/AbsAA_(48h) l_) × 100(3)

### 4.6. Evaluation of Anti-Inflammatory Activity

#### 4.6.1. Antidenaturation Assay

An antidenaturation test was carried out with the method reported by Trieu Ly et al. [78], with some modifications. The reaction mixture was constituted with 1 mL of extract or diclofenac sodium (500, 250, 125, 62.5, 31.25 and 15.625 µg/mL, prepared in distilled wate) and 1.9 mL of phosphate buffered saline (pH = 6.4) and 0.1 mL of fresh egg albumin. Then, the mixtures were incubated at 37 °C for 20 min, followed by heating at 70 °C for 5 min in a water bath. Finally, after cooling, the absorbances were measured with a spectrophotometer at 660 nm. The following formula was used to calculate the inhibition percentages of denaturation:Inhibition (%) = [(AbsC − AbsS)/AbsC] × 100(4)

AbsC is the absorbance of the control tube (distilled water), and AbsS is the absorbance of tubes containing extracts or diclofenac.

#### 4.6.2. Membrane Stabilization Test

The anti-inflammatory activities of the *Pentaclethra eetveldeana* leaves extracts were determined according to the method described by Trieu Ly et al. [78], with some modifications.

##### Erythrocyte Suspension

After collection from healthy volunteers in EDTA tubes, the blood was centrifuged (3000 rpm for 10 min) to remove the non-erythrocytic phase; then, the erythrocytes were washed via centrifugation using sodium phosphate buffer (pH = 7.4). After washing, the erythrocytes are taken up in an equivalent volume of physiological saline solution (0.9%). The suspension of erythrocytes was prepared at 40% (*v*/*v*) in isotonic buffer solution (0.2 g of NaH_2_PO_4_; 1.15 g of Na_2_HPO_4_; and 9 g of NaCl in 1 liter of distilled water). The study protocol was performed according to the Declaration of Helsinki and approved by the Health Sciences Research Ethics Committee (CERSSA) of the Republic of Congo (N° 442 March 2021).

##### Heat-Induced Hemolysis

A volume of 2 mL of plant extract or diclofenac (500, 250, 125, 62.5, 31.25 and 15.625 µg/mL, prepared in isotonic buffer solution) was mixed with 0.1 mL of erythrocyte suspension (40%), while 2 mL of saline solution (0.9%) was used as a negative control. After that, the mixtures were incubated in a water bath for 20 min at 56 °C. After incubation and cooling of the tubes at room temperature, the mixtures were centrifuged for 10 min at 2500 rpm. Then, the absorbances of the supernatants were measured at 540 nm.

##### Hypotonic-Induced Hemolysis

A volume of 2 mL of extract or diclofenac ((500, 250, 125, 62.5, 31.25 and 15.625 µg/mL, prepared in distilled water) was mixed with 0.1 mL of erythrocyte suspension (40%) in a test tube; then, the mixtures were incubated for 1 h at 37 °C. After incubation, centrifugation of the mixtures at 2500 rpm for 10 min followed. Finally, the hemoglobin content of supernatant was estimated at 540 nm. The following formula was used to calculate the inhibition percentages of hemolysis:Inhibition (%) = [(AbsC − AbsS)/AbsC] × 100(5)

AbsC is the absorbance of the control tube (distilled water) and AbsS is the absorbance of the tube containing the extract or diclofenac.

### 4.7. Statistical Analysis

Statistical analysis was performed using Microsoft Office Excel 2019 and SPSS. The results were expressed as mean ± standard deviation, and statistical significances between samples and the reference were denoted at the *p* < 0.05, *p* < 0.01 and *p* < 0.001 levels. The IC_50_ values were calculated for data obtained from the antioxidant activity. All quantifications of phenolic compounds and the antioxidant and anti-inflammatory tests were carried out in triplicate. Correlation analyses were conducted to determine any associations between the phytochemicals and antioxidant activity.

## 5. Conclusions

This study clearly showed that despite the differences in the collection regions of the *Pentaclethra eetveldeana* leaves, and except for flavonoids, there is homogeneity in their production of secondary metabolites. In addition, both flavonols and flavones are produced by *Pentaclethra eetveldeana* leaves; this shows heterogeneity in the types of flavonoids. Interestingly, from one region to another, secondary metabolites were produced proportionally, and this is the proof that the genotype of *Pentaclethra eetveldeana* is conserved for these four regions, even though the difference between regions influences secondary metabolites accumulation. Furthermore, the extracts demonstrated their ability to scavenge free radicals, stabilize the erythrocyte membrane and inhibit protein denaturation in vitro. Without a doubt, the secondary metabolites present are at the origin of the observed activities. All of these activities justify the use of *Pentaclethra eetveldeana* leaves in the traditional treatment of dementia. Moreover, they offer interesting leads in view of their activities at low concentrations. Obviously, aqueous extracts from *Pentaclethra eetveldeana* leaves have both antioxidant and anti-inflammatory activities. Finally, compared to diclofenac or ascorbic acid, these aqueous extracts offer an excellent perspective because whether it is diclofenac or ascorbic acid, only one activity is recognized for them.

## Figures and Tables

**Figure 1 plants-12-03271-f001:**
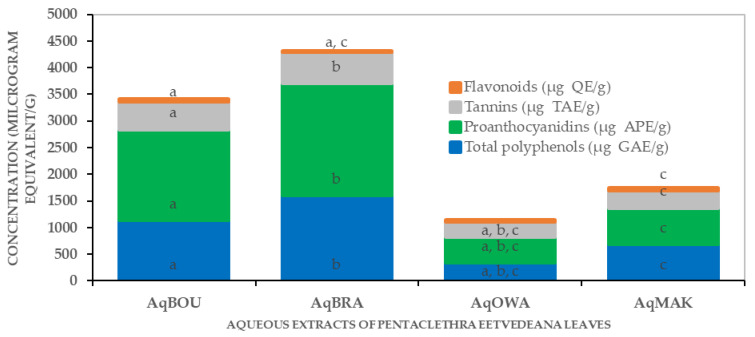
Total polyphenols, proanthocyanidins, tannins and flavonoids of *Pentaclethra eetveldeana* leaf extracts from Boundji, Brazzaville, Owando and Makoua, in units of µg of gallic acid equivalent (µg GAE), µg of apple proanthocyanidin equivalent (µg APE), µg of tannic acid equivalent (µg TAE), µg of quercetin equivalent (µg QE), respectively. For each compound, the same letter means a significant difference (*p* < 0.01) between the two.

**Figure 2 plants-12-03271-f002:**
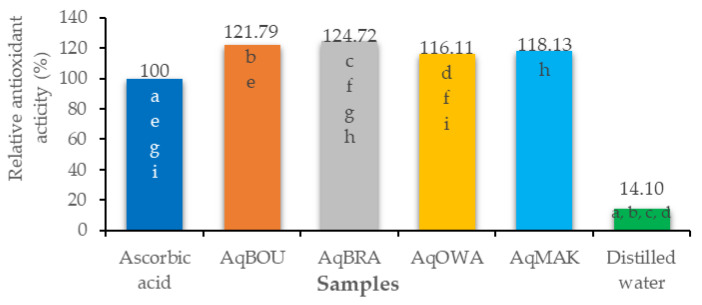
Relative antioxidant activities (%) of *Pentaclethra eetveldeana* leaf extracts. The same letter means significant difference (*p* ≤ 0.05).

**Figure 3 plants-12-03271-f003:**
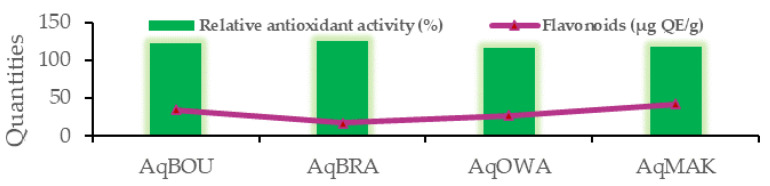
Positive correlations between relative antioxidant activities (%) and flavonoids (µg QE/g).

**Figure 4 plants-12-03271-f004:**
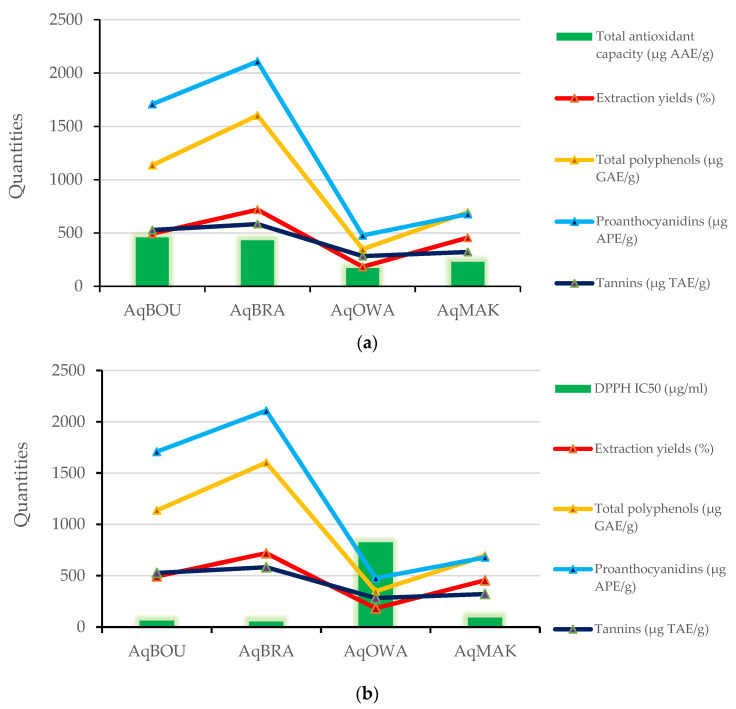
(**a**) Positive correlations between total antioxidant capacity (µg AAE/g), extraction yields (%), total polyphenols (µg GAE/g), proanthocyanidins (µg APE/g) and tannins (µg TAE/g). (**b**) Negative correlations between DPPH radical scavenging activity (µg/mL), extraction yields (%), total polyphenols (µg GAE/g), proanthocyanidins (µg APE/g) and tannins (µg TAE/g). IC_50_ and extraction yields were multiplied by 100 for better visibility in the figure.

**Figure 5 plants-12-03271-f005:**
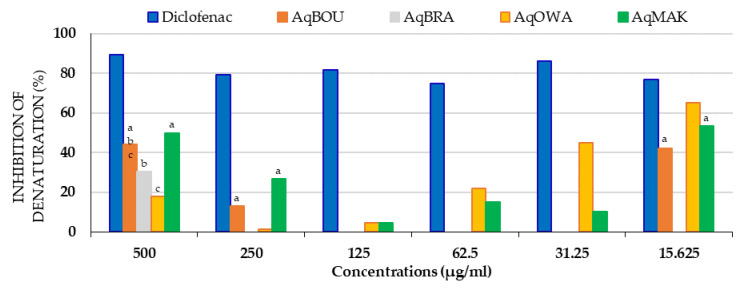
Antidenaturation activity of *Pentaclethra eetveldeana* leaf aqueous extracts. For the same concentration, the same letter means no significant difference; no letter in common or no letter means a significant difference (*p* < 0.05).

**Figure 6 plants-12-03271-f006:**
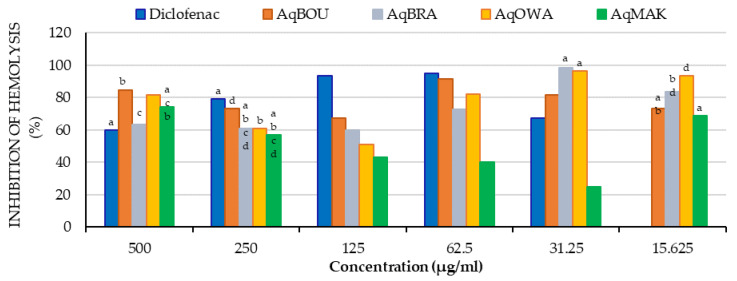
Inhibition (%) of heat-induced hemolysis by *Pentaclethra eetveldeana* leaf extracts. For the same concentration, the same letter means no significant difference; no letter in common or no letter means a significant difference (*p* < 0.05).

**Figure 7 plants-12-03271-f007:**
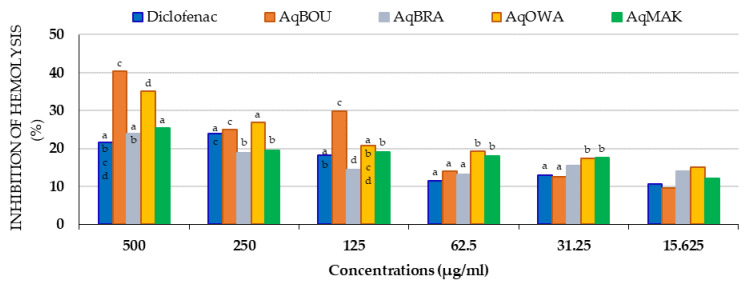
Inhibition (%) results of hypotonicity-induced hemolysis by *Pentaclethra eetveldeana* leaf extracts. For the same concentration, the same letter means no significant difference; no letter in common or no letter means a significant difference (*p* < 0.05).

**Figure 8 plants-12-03271-f008:**
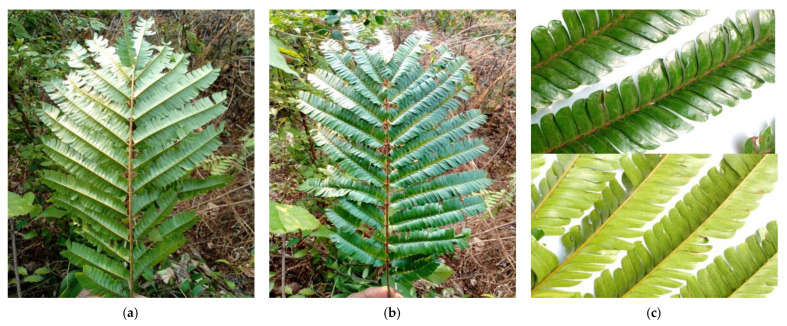
Leaves of *Pentaclethra eetveldeana*. (**a**) *Pentaclethra eetveldeana* leaves seen from behind; (**b**) *Pentaclethra eetveldeana* leaves seen from the front; (**c**) Close-up view of the sides of *Pentaclethra eetveldeana* leaves.

**Table 1 plants-12-03271-t001:** Yields of aqueous extracts of *Pentaclethra eetveldeana* leaves.

Extracts	Mass of Dry Extract (g)	Yields (%)
AqBOU	3.21	4.94
AqBRA	4.66	7.18
AqOWA	1.19	1.83
AqMAK	2.97	4.56

**Table 2 plants-12-03271-t002:** Phytochemical composition of *Pentaclethra eetveldeana* leaf aqueous extracts.

Extracts	Secondary and Primary Metabolites							
	Anthraquinones	Alkaloids	Cardiotonic Glycosides	Reducing Sugars	Polyphenols	Oses and Holosides	Sterols and Terpenes	Saponins
AqBOU	++	+++	++	+++	+++	+++	+	+++
AqBRA	+	++	++	+++	+++	+++	++	+++
AqOWA	+++	+	+++	+++	+++	-	-	+++
AqMAK	+++	++	+++	+++	+++	-	-	+++

Very positive reaction: +++; positive reaction: ++; weak reaction: +; negative reaction: -.

**Table 3 plants-12-03271-t003:** Phenolic compounds identified in *Pentaclethra eetveldeana* leaf aqueous extracts.

Extracts	Phenolic Compounds				
	Anthocyanins	Coumarins	Flavones	Flavonols	Gallic Tannins
AqBOU	-	+++	-	+++	+++
AqBRA	+	+++	-	+++	+++
AqOWA	-	+++	+++	-	+++
AqMAK	-	+++	+++	-	+++

Very positive reaction: +++; weak reaction: +; negative reaction: -.

**Table 4 plants-12-03271-t004:** Radical scavenging activity of extracts of *Pentaclethra eetveldeana* leaves.

Samples		DPPH IC_50_ (µg/mL)
Plant extracts	AqBOU	0.61 ± 0.11 ^a^
	AqBRA	0.53 ± 0.07 ^c^
	AqOWA	8.24 ± 1.32
	AqMAK	0.91 ± 0.13 ^a,c^
Ascorbic acid		10.27 ± 0.27

Values are expressed as means ± standard deviation (SD). The same letter means a significant difference between the two. ^a,c,^ *p* = 0.05.

**Table 5 plants-12-03271-t005:** Total antioxidant capacity of extracts of Pentaclethra eetveldeana leaves.

Extracts	Total Antioxidant Capacity (µg AAE/g)
AqBOU	457.04 ± 11.25 ^a^
AqBRA	431.62 ± 12.99 ^b^
AqOWA	169.95± 18.18 ^a,b,c^
AqMAK	229.54 ± 8.53 ^c^

Values are expressed as means ± standard deviation (SD). The same letter means a significant difference between the two. ^a,b^ *p* = 0.001 and ^c^
*p* = 0.05.

**Table 6 plants-12-03271-t006:** Correlation coefficients.

	Extraction Yields	TP	PR	TN	FL
Total polyphenols (TP)	0.956	1			
Proanthocyanidins (PR)	0.872	0.974	1		
Tannins (TN)	0.858	0.965	0.998	1	
Flavonoids (FL)	−0.345	−0.510	−0.535	−0.501	1
DPPH IC_50_	−0.866	−0.758	−0.678	−0.687	−0.158
Total antioxidant capacity	0.796	0.903	0.958	0.972	−0.314
Relative antioxidant activity	−0.255	−0.305	−0.241	−0.191	0.886

**Table 7 plants-12-03271-t007:** Tests for identification of metabolites in extracts of *Pentaclethra eetveldeana* leaves.

Metabolites	Tests
Polyphenols	The reaction is positive for a bluish black color after the addition of a few drops of FeCl_3_ to the extract [79].
Tannins	After addition of a few drops of FeCl_3_ (5%) to the extract, the reaction is positive for gallic tannins for a green color, while for the pseudo tannins, a brown color is observed [80,81].
Alkaloids	The reaction is positive for the observation of an orange precipitate after the addition of a few drops of Dragendroff’s reagent to the extract [80].
Flavonoids	The reaction is positive for flavonols on observation of a red color after the addition of hydrochloric alcohol (HCl/ethanol, 50:50, *v*/*v*) to the extract followed by 5 to 6 magnesium shavings. That of flavones gives an orange color [81,82].
Saponins	The reaction is positive when an abundant and persistent foam is observed for more than one minute after vigorous shaking of the extract [83].
Cardiotonic glycosides	The reaction is positive when a reddish-brown color is observed after the addition of 2 mL of chloroform followed by 2 mL of sulfuric acid to the extract [84].
Reducing sugars	The observation of a brick-red precipitate after the addition of 1 mL of Fehling’s liquor to the extract indicates the presence of reducing sugars [85].
Oses and holosides	The reaction is positive when a red color is observed by adding a few drops of sulfuric acid to the extract, followed by few drops of ethanol saturated with thymol after 5 min [86].
Sterols and triterpenes	2 mL of chloroform followed by 2 mL of concentrated sulphuric acid are carefully added from sides of the test tube. The appearance of a red ring is the positive result for sterols, while a reddish-brown coloration indicates the presence of triterpenoids [87].
Coumarins	The observation of yellow, red, green, blue or violet fluorescence under a UV lamp indicates the presence of coumarins [88].
Anthraquinones	The reaction is positive for a red color after the addition of 1 mL of aqueous ammonia solution (25%) to the extract [89].
Anthocyanins	The reaction is positive when a pink-red color which turns blue-purple is observed after the addition of 2 mL of hydrochloric acid (2 N) followed by 1 mL of an aqueous solution of ammonia (25%) [90].

## Data Availability

Not applicable.

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
