# Peer review of "Pentaclethra eetveldeana* Leaves from Four Congo-Brazzaville Regions: Antioxidant Capacity, Anti-Inflammatory Activity and Proportional Accumulation of Phytochemicals"

_plants, 2023, doi:10.3390/plants12183271_

Round 1
Reviewer 1 Report
The authors of the article study the antioxidant and anti-inflammatory activities of Pentaclethra eetveldeana aqueous extract, and also explain the possible reason for the use of the plant in folk medicine for the treatment of dementia. There are a number of clarifications and questions to the materials of the article:
1. In the legend for tables 2 and 3, there is no decoding of the "-" symbol, which should be attributed to "negative reaction"
2. The authors should try to explain such a large difference in the accumulation of secondary metabolites by the same species living in different ecological conditions. What factors could have had the greatest impact on the synthesis and accumulation of metabolites
3. According to the authors, the plant retains a proportional synthesis of metabolites, however, judging by Table 1, the population with the lowest yield of extractives showed an order of magnitude higher antioxidant activity in the DPPH test, what is the reason?
4. The ratio between and within the classes of synthesized metabolites will depend to a greater extent on epigenetic factors than on the genotype, since this is an adaptive response of the plant organism to environmental factors
Author Response
We have taken your questions and suggestions into account and have provided answers and corrections as specified in the attached file.
Reviewer 2 Report
The authors aim to evaluate antioxidant and anti-inflammatory activities as well as the phytochemical composition of endemic plant Pentaclethra eetveldeana leaves from four Congo-Brazzaville regions.
After investigation, plant extracts showed higher antiradical activity and inhibition of lipid peroxidation than ascorbic acid, membrane stabilizing power of aqueous extracts was superior or comparable to diclofenac while the same extracts were less effective for inhibition of denaturation. All aqueous extracts contained polyphenols, saponins, alkaloids, anthraquinones, reducing sugar and cardiotonic glycosides. Total polyphenols, tannins and proanthocyanidins are produced proportionally from one region to another. Finally, leaves from Brazzaville and Boundji contained flavonols while those from Makoua and Owando contain flavones. Thus, the authors concluded that Pentaclethra eetveldeana leaves contribute in dementia traditional treatment through their antioxidant and anti-inflammatory properties.
The manuscript is well written, well structured, and data are clearly presented. The observation is: the data are only preliminary; all the methods used are usual and although they are reliable, studies must also be done at the molecular level. In addition, the data are, at this level, only of local interest. The authors must continue the investigations further. I think that the anti-dementia effect cannot be so easily concluded from a good antioxidant and anti-inflammatory activity.
Small observations:
- In Table 3, replace Makoua with AqMAC
- There are passages in which data from the tables are repeated in the text (ex: lines 186-191)
- Sometimes could be replaced the text with a table (ex: lines 166-176), such as the results could be analyzed better
Please check the spelling and grammar in all the manuscript, small mistakes are occurred:
As an example, in Introduction, line 39:
…while the same extracts was less effective for inhibition of denaturation
Author Response
We have considered the author's remarks and suggestions. we reviewed the analyzes as well as the spelling and grammar throughout the manuscript.
